# An Order-Preserving Encryption Scheme Based on Weighted Random Interval Division for Ciphertext Comparison in Wearable Systems

**DOI:** 10.3390/s22207950

**Published:** 2022-10-18

**Authors:** Ruowei Gui, Liu Yang, Xiaolin Gui

**Affiliations:** 1Faculty of Electronic and Information Engineering, Xi’an Jiaotong University, Xi’an 710049, China; 2Shaanxi Provincial Key Laboratory of Computer Network, Xi’an Jiaotong University, Xi’an 710049, China

**Keywords:** order-preserving encryption, random interval division, privacy protection, wearable devices, IoT

## Abstract

With the rapid development of wearable devices with various sensors, massive sensing data for health management have been generated. This causes a potential revolution in medical treatments, diagnosis, and prediction. However, due to the privacy risks of health data aggregation, data comparative analysis under privacy protection faces challenges. Order-preserving encryption is an effective scheme to achieve private data retrieval and comparison, but the existing order-preserving encryption algorithms are mainly aimed at either integer data or single characters. It is urgent to build a lightweight order-preserving encryption scheme that supports multiple types of data such as integer, floating number, and string. In view of the above problems, this paper proposes an order-preserving encryption scheme (WRID-OPES) based on weighted random interval division (WRID). WRID-OPES converts all kinds of data into hexadecimal number strings and calculates the frequency and weight of each hexadecimal number. The plaintext digital string is blocked and recombined, and each block is encrypted using WRID algorithm according to the weight of each hexadecimal digit. Our schemes can realize the order-preserving encryption of multiple types of data and achieve indistinguishability under ordered selection plaintext attack (IND-OCPA) security in static data sets. Security analysis and experiments show that our scheme can resist attacks using exhaustive methods and statistical methods and has linear encryption time and small ciphertext expansion rate.

## 1. Introduction

The popularity and wide application of wearable devices are improving many different fields, such as medical care, crowdsourcing services, and IoT applications. In particular it enables multiple physical objects to sense, process, and exchange health data transparently and seamlessly through the network. These health data explode and distribute in multiple data gateways and healthcare centers. Integrating health data from these multiple sources to solve the problem of information islands is crucial for efficient data release, utilization, and analysis [1]. If distributed health data can be effectively aggregated, multiple valuable healthcare-related services can be obtained. Various data users, such as governments and researchers, can model or count aggregated data. For example, the maximum, minimum, and average blood pressure in a specific population can be studied in public health. Although data aggregation in cloud can provide important medical services for the development of public medical management, due to the openness of the network and the privacy sensitivity of health data, security threats and privacy leaks may occur in the process of data aggregation [2].

Encryption is an effective solution to protect data privacy. However, when encrypted data are aggregated to the cloud, data comparison becomes difficult or even impossible. The main reason is that the traditional encryption scheme does not support sorting encrypted data. In this case, in order to perform sorting the encrypted data must be decrypted first. The multiple encryption and decryption processes of data significantly increase the overhead of private data processing. In order to balance security and availability, a practical method is to design a new encryption scheme, which supports ranking, ordering, comparison, and range queries over encrypted data (called ciphertext).

Order-preserving encryption scheme (OPES) [3,4] is also an effective scheme to support ciphertext comparison. In this scheme, the ciphertext preserves the natural ordering of the plaintexts, that is, if the x and y are known plaintexts and the encryption function is Enc (plaintext), then when x > y, there is Enc (x) > Enc (y). The OPES allows sorting, comparison, and range query of encrypted data on an untrusted outsourced server without requiring the server to obtain a decryption key. For example, in the health sensing system, in order to query the range of blood pressure (BP) (e.g., 60 < BP < 90), the server only needs to select all encrypted data between Enc (60) and Enc (90). Obviously, the OPES not only guarantees the data privacy, but also improves the efficiency of data query because the range query can be performed without decrypting the ciphertext.

Order-preserving encryption enables efficient range queries on the encrypted data, but the security is debatable. Indistinguishability against chosen plaintext attack (IND-CPA) is used to illustrate the security strength of traditional encryption algorithms. However, because the ciphertext preserves the order of plaintext, OPE cannot meet the traditional security strength of IND-CPA. Boldyreva [5] first proposed a security notion for OPES called indistinguishability under ordered chosen-plaintext attack (IND-OCPA).

After IND-OCPA, some other more relaxed OPES security notions were proposed. These security notions are weakened versions of IND-OCPA and thus can leak more information than just ordering.

### 1.1. Related Works

Agrawal [3] first proposed an order-preserving encryption scheme (OPES) for numeric data in databases, this encryption scheme allows efficient range queries on encrypted data. OPES maps the plain text to ciphertext by constructing piecewise monotone functions.

Boldyreva [4] proposed a practical OPES named LazySample, which is based on natural relation and hyper geometric probability distribution. LazySample divides the ciphertext space into two non-intersect sub-spaces according to the hyper geometric probability distribution, and then mapped the plain text into one of these sub-spaces by value.

Boldyreva [5] defined the security notion IND-OCPA where the adversary can only query the left-or-right encryption oracle with ordered plaintext pairs. An encryption scheme is secure under IND-OCPA if the advantage of an efficient adversary (probability to distinguish whether the returned ciphertexts are encrypted from the left or the right plaintexts) is negligible. IND-OCPA is the highest security notion (with respect to indistinguishability and left-or-right encryption oracle) for OPES. However, it can be shown that the OPES is susceptible to the big jump attack and cannot be secure under IND-OCPA unless its ciphertext-space is exponential in the size of the plaintext-space.

Kadhem [6] presented a database encryption scheme called MV-POPES, which allowed privacy-preserving queries over encrypted databases with an improved security level. The MV-POPES divides the plain text domain into many partitions and randomize them in the encrypted domain.

Liu [7] proposed an order-preserving scheme based on nonlinear function for indexing encrypted data, facilitating the range queries over encrypted databases. Yum [8] proposed the OPE to utilize the non-uniform distribution of the plaintext space to reduce the number of sampling algorithm invocations.

Popa [9] first proposed a stateful ideal secure OPE scheme satisfying IND-OCPA. Compared with Boldyreva’s stateless scheme, this scheme needs to use a tree to track the order distribution between all plaintext and ciphertext. The core idea of mOPE is to use the balanced binary search tree to encode data into ciphertext. The ciphertext is stored into an untrusted database. When a trusted client inserts new data into the database, the balance of the original binary search tree may be broken, which requires readjusting the binary search tree and obtaining the insertion position of the new data using an interaction protocol. Obviously, the dynamic construction of the tree greatly increases the computation and communication costs.

Seungmin [10] proposed an order-preserving encryption scheme based on series expansion, which is named chaotic order-preserving encryption (COPE). The core idea of COPE was to convert the plaintext into the coordinate space which was constructed according to the key.

Martinez [11] proposed an order-preserving symmetric encryption scheme whose encryption function is recursively constructed. This scheme starts from the trivial order-preserving encryption function, which is the identity, a function is constructed in a series of steps by making it more and more complex until the desired security level is reached.

Lee [12] proposed a group order-preserving encryption scheme (GOPES) to support query processing over encrypted data. This scheme can preserve the order of each data group by generating the signatures of the encrypted data, it can provide a high degree of data privacy protection both order matching attacks and data count attacks.

Quan [13] proposed a mutable top OPES (TOPES) to enable top-k queries on encrypted data with minimized information leakage. With TOPES, the ciphertexts of top-k values are the top-k in the ciphertext domain, while the ciphertexts of non-top-k values are in meaningless order.

Liu [14] proposed a new order-preserving encryption model, which uses message space expansion and nonlinear space split to hide data distribution and frequency, and further analyze its security against two kinds of attack including ciphertext-only attack and statistical attack.

Kerschbaum [15] presented a secure OPES: randomize the ciphertexts to hide the frequency of plaintexts. Yang [16] proposed a semi-order-preserving encryption scheme to guard against the differential attack. Li [17] presented the security analysis about one-to-many OPES cloud data search. Kim [18] constructed an efficient and programmable OPES for outsourced databases and commented on the order-preserving encryption security schemes using pseudo-random function.

Shen et al. in [19] proposed a practical and secure stateless order-preserving encryption scheme. This scheme can achieve the indistinguishability under committed ordered chosen plaintext attacks (IND-CCPA) security for static data set. The IND-CCPA security cannot be met for dynamic data set, and is a constrained version of IND-OCPA.

### 1.2. Our Contributions

According to the above literatures, we found that most of the existing OPES focus on the numeric data in databases. However, OPES is being extended to the character string data (CSD), rather than confining to the numeric data (ND). Further, most of existing OPES have some key limitations that need to be addressed, for example:(1)Some existing OPES usually convert each character into the corresponding ASCII code while processing CSD. With such a simple method, the attackers can easily deduce the mapping between each pair of plaintext and ciphertext. For example, the most frequently used 52 English characters have only 52 ASCII codes; therefore, only by using the exhaustive method, the attackers can obtain the code book;(2)Some existing OPES encrypt each character into a float value, and then convert the float value to hexadecimal string or byte array with fixed size; therefore, the scheme is vulnerable to the size-based attacks. Once the size of ciphertext is obtained, attackers can easily deduce the size of plaintext and the content of the plaintext as well;(3)Most existing OPES are determined encryption schemes with one-to-one mapping between the ciphertext and plaintext. Moreover, how to balance the security and computation efficiency also need more attention. For example, to improve the security, the domain size should be increased; however, when the domain size is increased, the computation efficiency declines exponentially.

For addressing the above problems, we propose a novel order-preserving encryption scheme based on weighted random interval division (WRID-OPES) in this paper. This scheme is further extended to support multiple data types, such as integers, floating numbers, and strings scenarios.

The main contributions of this paper are summarized as follows:(1)We designed a new character encoding method according to the occurrence frequency in preprocess phase. Based on this method, the CSD is converted into ND based on Basic Multilingual Plane ((BMP) Unicode from U+0000 to U+FFFF);(2)We proposed a weighted random interval division algorithm (WRID). This algorithm guarantees that the partition ratio of each interval is random and that the distribution of division ratios is closed to the predetermined weight value. The division ratio refers to the size of the divided sub-intervals;(3)We constructed an order-preserving random interval tree (OPRIT) based on WRID. Based on OPRIT, we implemented the numeric string encryption scheme, which divides the numeric string into segments and maps each segment into the cryptogram space and can achieve the IND-OCPA security strength.

The remainder of this paper is organized as follows: Section 2 provides our designed solution of OPES. Section 2.1 describes the preprocess method of character strings, Section 2.2 proposes an order-preserving encryption scheme based on random interval division (WRID), Section 2.3 introduces the order-preserving random interval tree (OPRIT), Section 2.4 shows the detail of the WRID-OPES algorithms. In Section 3, we conduct a series of experiments to evaluate the WRID-OPES. Section 4 provides the conclusion and further works.

## 2. Our Proposed Solution

In this section, we describe the design principle of WRID-OPES, including the pre-process method of character strings, the weighted random interval division algorithm (WRID), the order-preserving random interval tree (OPRIT), and the encryption process using OPRIT.

### 2.1. The Pre-Process Method of Character Strings

In WRID-OPES, the character string (plaintext) needs to be encoded into numeric strings. The encoding process should conform to two conditions:(1)The first one is order-preserving, which means that the converted data should preserve the order of original data;(2)The second one is reversibility, so that we are able to retrieve the original data according to the converted data.

At present, various character sets are used in computers, but most of them cannot satisfy both of the above two conditions at the same time. In this paper, we choose the UTF-16BE in the basic multilingual plane (BMP, code points U+0000 to U+FFFF) as the coding foundation; however, the using of UTF-16BE also causes vulnerability. Because UTF-16BE is a fixed coding approach, all the characters are also converted into hexadecimal strings with the same size. The converted data are regular in some ways, especially if the plaintext strings are mainly composed of numbers and alphabetic characters, then the converted codes always start with two continuous ‘0’, followed by their ASCII codes. Therefore, we made some modifications to the original UTF-16BE. The modified encoding meets the previous two conditions as before. Meanwhile, the problem of periodic padding the same character is addressed.

The changed UTF-16BE is shown in Table 1. We mainly changed it from four aspects:(1)Codes from “0000” to “007F” cover the basic Latin characters, including the English characters, Arabic numerals, and symbols. They are frequently used in the English texts. At the same time, the prefix “00” makes converted ND to be periodic. Therefore, we deleted the same prefix and use their ASCII code as the change code;(2)Codes between “0080” to “3FFF” cover the native characters and other types of characters. The native characters include the expanded Latin characters, Greek characters, Hebrew characters, Arabic characters, etc. Other types of characters include mathematical operators, square elements, etc. These characters often appear in specialized texts only. So, in this paper, we added a prefix “8” to preserve the order;(3)CJK characters refer to the characters in Chinese, Japanese, and Korean, whose occurrence frequency is high. In this paper, we did not add any additional prefix for these characters but transform them to the interval between “9000” and “EFFF” to preserve order as well;(4)The code between “A000” to “FFFF” refers to other East Asian characters, diacritics characters, and reserved characters. We added a prefix “F” to the original code to preserve order.

In the preprocess method, in order to obtain the occurrence frequency of each numerical character, we count the occurrence times of each numerical character in the converted numerical string, then count the occurrence frequency and weight (*W*) for each numerical character.

### 2.2. The Weighted Random Interval Division Algorithm (WRID)

In this section, we proposed the weighted random interval division algorithm (WRID) based on the encoded numeric strings. The WRID must meet two objectives:(1)For a single interval division, each interval must be randomly divided in WRID;(2)After multiple interval divisions, the expectation of division percentage for each interval must be close to the predetermined weight value.

Here, we introduce the definition of weighted random interval division, as shown in Definition 1.

**Definition** **1:**
*The weighted random interval division.*


Provided a set of weight value *W = {w_0_, w_1_,…w_n_}*, where *w_k_* ∈ (0,1) and $\overset{n}{\underset{k=1}{\sum }}w_{k}=1$, if *X* = {*x_0_, x_1_,…x_n_*} meets the following conditions, ***X*** is defined as a weighted random interval division in [0,1) and ***x_i_*** is the length of the i-th weighted random sub-interval.

Condition 1: ***x_i_*** is a random number in interval (0,1);

Condition 2: $\overset{n}{\underset{k=1}{\sum }}x_{k}=1$;

Condition 3: $E\left(x_{k}\right)=w_{k}$.

where *k* = 0,…, n, and $E\left(x_{k}\right)$ is the expectation of $x_{k}$. Obviously, with Definition 1, we can divide intervals randomly into sub-intervals while the statistical results for the random sub-intervals are approximate to statistical results for the sub-intervals divided using fixed ratio.

**Definition** **2:**
*The random weighted tree.*


The random weighted tree is a binary tree that the weight of any non-leaf node is the sum of the weight of all its children nodes, while the weight of the root node is the sum of the weight of all the leaf nodes. Figure 1a is an example of a weighted tree with seven leaf nodes.

Utilizing the correspondence between the weighted tree and interval division, we are able to divide the original interval progressively into several weighted random sub-intervals. Figure 1b provides the interval division result corresponds to the tree shown in Figure 1a.

From this example, we can see that:(1)The number of leaf nodes in the weighted tree equals the number of sub-intervals;(2)The weight of the i-th leaf node equals to the proportion of the i-th sub-interval in the original interval when dividing the interval with fixed ratio.

#### 2.2.1. The Design of WRID Algorithm

In this section, we recursively construct the weighted tree from the bottom to up. For a given group of weight ***W*** and the number of sub-intervals ***n***, we can first create a set ***S***, which contains ***n*** trees. Every tree in set ***S*** has and only has a single root node with weight ***w_i_***. Where ***w_i_*** is the *i**-th* value in ***W***.

The following steps recursively merge the trees in set ***S*** and finally output a complete weighted tree:

Step 1. If there is only one tree ***T*** in set ***S***; Then ***T*** is the weighted tree, and return ***T*** and suspend; otherwise, go to Step 2;

Step 2. ***S*** is divided into two subsets with the same size. Two subsets respectively are named as left sub-set ***S_left_*** and right sub-set ***S_right_***;

Step3. For each of the two subsets, repeat above two steps until ***S_left_*** is merged to a single tree ***T_left_*** and ***S_right_*** is merged to tree ***T_right_***;

Step4. A new tree ***T*** is created, whose left sub-tree and right sub-tree are respectively ***T_left_*** and ***T_right_***. Let the weight of root node of ***T*** is the sum of the weight of root nodes of ***T_left_*** and ***T_right_***;

Step5. Output ***T***.

Based on Definitions 1 and 2, referring to the above steps, we construct the WRID algorithm which is shown in Algorithm 1.

Algorithm 1 includes three input parameters, which are the input interval I=[α, β), initial the seed ***sd*** of key, and the sub-interval sequence number ***m***, respectively. The algorithm outputs the ***m***-th weighted random sub-interval ***I_m_***.
**Algorithm 1:** The weighted random interval division algorithm (WRID)InputOutput***I = [α, β), sd, m******I_m_***Step1Initial node ***n*** as the root node of the weighted tree, assign ***nextLeft (sd)*** to ***sdl*** and ***nextRight (sd)*** to ***sdr***.Step2If ***n*** is leaf node, assign ***[α,β)*** to ***I_m_*** and exit this algorithm; otherwise, go to step 3Step3For the node ***n*** (weight is w), its children nodes are ***nl*** (left child node, weight is ***wl***) and ***nr*** (right child node, weight is ***wr***). Constructing the independent uniform random variable ***Rl (sdl)*** and ***Rr (sd******r)*** in the domain [0,1). Then computing ***x = (wl * rl)/(wl * rl + wr * rr)***, where ***rl*** and ***rr*** are two random number generated according to ***Rl (sdl)*** and ***R******r (sd******r)***.Step4If the ***m******-th*** leaf node of weighted tree is in the left children tree of ***n***, then computing as follows: ***β***
***=***
***α***
***+***
***(β-α) * x,***
***n***
***=***
***nl, sdl***
***=***
***nextleft (sdl), sdr***
***=***
***nextLeft (sdr)***. Then go to step 2.Step5If the m-th leaf node of weighted tree is in the right children tree of ***n***, then computing as follows: ***α***
***=***
***α***
***+***
***(β-α)***
*******
***x, n***
***=***
***nr, sdl***
***=***
***nextRight (sdl), sdr***
***=***
***nextRight (sdr).*** Then go to step 2.

In Algorithm 1, we used an iterative approach to compute the weighted random sub-interval ***I_m_*** according to the weighted tree. Functions ***nextLeft (******sd)*** and ***nextRight (******sd)*** are used to generate new seeds in each iteration of the division based on the current seed to improve the algorithm security.

Algorithm 1 provides us a way to divide a complete interval randomly into several sub-intervals, and all these sub-intervals comply with the given distribution. By using mathematical induction, we can prove the correctness of Algorithm 1. That is, for any given interval ***I = [α, β)***, we can obtain its ***m-th*** sub-interval ***I******_m_ = [α******_m_, β******_m_)***, which satisfies equation: E***(******β******_m_***
***− α******_m_******) = (β***
***−α)***
**** w_m_***. Here, ***w_m_*** is the weight of the m-th sub-interval.

#### 2.2.2. The Randomness of Intervals in WRID

In order to meet the order-preserving request, a termination character *λ* is added to the end of every converted numerical string. Therefore, numerical strings include 17 different kinds of symbols: *λ* and numerical character 0 to F.

We used the correlation coefficient (*R*) and the coefficient of variation (*V_σ_*) to measure the randomness of divided sub-intervals of the same input interval but different seeds.

**Definition** **3.**
*Correlation coefficient (R).*


*X* and *Y* are two random variables. The correlation coefficient between *X* and *Y* is defined as the following:(1)$$R=\frac{Cov\left(X,Y\right)}{\sqrt{D\left(X\right)}\;\sqrt{D\left(Y\right)}\;\;}$$
where *Cov(X,Y)* is covariance between ***X*** and ***Y***, which is defined as ***E([X − E(X)] * [Y − E(Y)])***. ***D(X)*** is the variance of ***X***.

The *R* is a measurement of the strength and direction of the linear relationship between two random variables. If ***|R|*** is close to 1, the relationship between the variables is stronger. Moreover, if *|R|* is 0, it means that the variables were independent.

**Definition** **4.**
*Variation Coefficient (V_σ_).*


*X* is a random variable, *σ* is the standard deviation of *X*. The variation coefficient for *X* is defined as follows:(2)$$V_{\sigma }=\frac{\sigma }{E\left(X\right)}$$
where $\sigma =\sqrt{{\left|X-E\left(X\right)\right|}^{2}}$ and *E(X)* is the expectation of *X*.

The variation coefficient is a normalized measurement of dispersion of a probability distribution or frequency distribution.

For testing the randomness of intervals, we generated the 1000 test samples, the length of each sample string was 20 and the definition domain was set to [0, 10^10^). 

Based on these samples, we used the following steps to perform the correlation test:(1)Compute the weight (*W*) according to the generated samples;(2)Divide the definition domain repeatedly for 1000 times by using WRID with two different seeds, and taking the lower boundary value of each sub-interval as the random variable;(3)Calculate R composed by the lower boundary values.

Based on same samples, we used the following steps to perform the *V_σ_* tests:(1)Divide the definition domain repeatedly for 1000 times by using WRID, and taking the length of each sub-interval as the random variable;(2)Calculate *V_σ_* of the set composed by the length of each sub-interval.

The experimental results are shown in Table 2. In this table, the ID of the sub-interval corresponds to the characters λ and 0-F.

From this table, we can easily observe that the largest *R* was 0.0583, which is closed to 0 and means that the boundary values of the divided sub-intervals were almost independent. On the other hand, we can observe that the largest value of coefficient of variation was 200.96%, and the smallest value was 88.03%. There are only three sub-intervals whose *V_σ_* is slighter smaller than 1. The *V_σ_* for other sub-intervals are all larger than 1. The results show that the lengths of the sub-intervals are very different from each other. Therefore, we can conclude that the relationship between the sub-intervals is weak by using the WRID. Moreover, the interval division is disperse and stochastic.

#### 2.2.3. The Expectations of Interval Division Percentage in WRID

For any given interval ***I*** and seed ***sd***, The WRID algorithm divides the given interval into 17 sub-intervals every time.

In this section, we randomly divide the definition domain for 1000 times by using different seeds each time. Then we can obtain the proportion for each sub-interval. Finally, we obtain the expectation value of the division ratios.

In Figure 2, we show the experimental results, which verifies that the expectation of proportion of sub-intervals divided by WRID algorithm in the entire domains closed to the pre-determined weight value ***W***. Therefore, WRID algorithm can realize the function of weighting.

### 2.3. The Encryption of Numerical String Based on WRID and OPRIT

In this section, we first construct an order-preserving random interval tree (OPRIT) based on Han’s researches in [20] and WRID algorithm, and then dynamically map the M-ary encoded numerical data (plaintexts space) to the ciphertexts space by dividing them into data segments using OPRIT and WRID.

#### 2.3.1. The Order-Preserving Random Interval Tree (OPRIT)

In order to construct an order-preserving random interval tree, we first provide the definition of the OPRIT and its related properties.

**Definition** **5.**
*An M-ary OPRIT is a multi-way tree which satisfies the following properties:*


**Property** **1.**
*Each non-leaf node in OPRIT has M + 1 ordered child nodes, are the **λ** node, and **M** value nodes from left to right. The **λ** node is a special leaf node, which refers to the end of the whole numerical string. Moreover, the value of the **λ** node is always the smallest among all the child nodes with the same parent node;*


**Property** **2.**
*Each node in OPRIT represents a continuous interval [α, β); The i-th child node represent the i-th sub-interval of the interval of the parent node, here, we define it as [αi, βi);*


**Property** **3.***The interval set of all the child nodes are a partition of the interval of parent node:*$\cup_{k=0}^{M}\left[\alpha_{k}-\beta_{k}\right)$ = [α−β). *If the interval of a child node is covered by any of the pre-divided target interval, then the child node is a leaf node, we call this kind of node an indivisible value node;*

**Property** **4.**
*We used WRID algorithm to divide each node into child nodes. The seed used by WRID is decided by the security key and the values of all the ancestor nodes of the current node;*


**Property** **5.**
*There are only two kinds of leaf nodes in OPRIT: λ node and indivisible value node. When a node is calculated as a leaf node, the algorithm output the corresponding interval for this leaf node as the value range for the ciphertext snippet. If the leaf node is not the λ node, the algorithm continues executing from the root to calculate another OPRIT, or the algorithm terminates;*


**Property** **6.**
*OPRIT is a virtual tree, which means that we only compute a path from the root node to the leaf node for each OPRIT rather than all the nodes in the whole tree.*


In Figure 3, we illustrate an example of OPRIT and the corresponding interval division results when *M* = 3. Figure 3a shows the encryption path to process the ternary numerical string “00”, “020”, “021”, “20”, and “21”, where circle node stands for the non-leaf node, the triangle node stands for the *λ* node and square node stands for the leaf node. Figure 3b shows the corresponding interval division results.

#### 2.3.2. The Encryption of Numerical String Using OPRIT

In this section, we first introduce the encryption keys of OPRIT. Then, we detail the encryption processes based on WRID algorithm when using OPRIT.

(1)The encryption keys of WRID-OPES

The encryption key of WRID-OPES consists of three parts, which are user password: *Userkey*, definition domain: *DD*, and target bucket number: *N*. Among them, the latter two parts are optional to users. When their values are not provided, the default value is used.

(2)The encryption processes of numerical string using OPRIT

Using OPRIT, we designed a numerical digital string encryption algorithm based on WRID, as shown in Algorithm 2. In this algorithm, ***Seedp*** is generated using seed (***Userkey***) function for first character, the next ***Seedp*** is generated according to the current ***seed******p*** and the numerical string ***ns*** converted from characters, e.g., ***Seedp*** = seed (***Seedp***, ***ns***).
**Algorithm 2:** The encryption processes of numerical string using OPRIT and WRID**Input****Output***Userkey*, *DD**, N**, ns* Ciphertext string: *c_i_*Step1Dividing uniformly the definition domain ***DD = [min,max)*** into ***N*** non-intersect buckets ***J(1),…J(k),…J(N)***. The size of each bucket is ***(max-min)/N***;Step2Adding the termination symbol ***λ*** in the last of numerical string ***ns*** (e.g., ***ns + λ******→******ns***). Then initial ci into a null string.Step3Initialing the root node interval and seed: *I **= DD, seedp*** = seed (***Userkey***).Step4Obtaining the first unprocessed numerical character m from ns, where *m**∈**{λ, 0, … 9, A,…,F}*. Step5Using *I_m_* = WRID (***I, seedp***) to obtain *I*_m_, and assign *I_m_* to *I*, e.g., *I*_m_***→******I***.Step6Computing the ciphertext ***r*** from *I_m_* using *seedp*, and converting the ***r*** to a numerical string ***r’*** and appended to the last of ciphertext ***c_i_***, (e.g., *c_i_ + r’****→****c_i_*).Step7If there is *I*⊆***J(k) (k = 1,…,N)***, which means that the algorithm is processing the undivided value node, then, go to step 3.Step8If there is ***m = λ***, it means that the end of ***ns*** has reached, then, output ***c_i_*** and exit the algorithm.Step9Computing the new seed ***seedp*** for the next division using *m* and *seedp*, e.g., ***seedp*** = Seed (***seedp, m***). Go to step 4

**Theorem** **1.**
*The Algorithm 2 is order-preserving. That is, for the numerical string segment ns_1_ and ns_2_ and the ciphertext **c_1_** and **c_1_**, if **ns_1_ < ns_2_**, then **c_1_ < c_2_**.*


**Proof.** We used *a_i_* to denote the ***i-th*** numerical character in ***ns, ns(i,j)*** to denote the sub-string of ns from index ***i*** to index ***j***, ***ns(i)*** to denote the sub-string from index ***i*** to the end of the ns, and ***I(ns)*** to denote the sub-interval obtained by OPRIT. We consider the following three scenarios. □

For the two numerical strings who have different prefixes, which are $ns_{1}=a_{1}ns_{1}^{\left(2\right)}$ and $ns_{2}=a_{2}ns_{2}^{\left(2\right)}$. We assume that ***a_1_***
***<***
***a_2_***. According to the definition of OPRIT, we know that all the values in ***I(a_1_)*** are smaller than the value in ***I(a_2_)*** and $I\;\left(a_{1}ns_{1}^{\left(2\right)}\right)\subset I\left(a_{1}\right)$*,* $I\;\left(a_{2}ns_{2}^{\left(2\right)}\right)\subset I\;\left(a_{2}\right)$; therefore, we can obtain that all the value in $I\;\left(a_{1}ns_{1}^{\left(2\right)}\right)$ are smaller than the value in $I\;\left(a_{2}ns_{2}^{\left(2\right)}\right)$. Then $\forall c_{1}\in I\left(a_{1}ns_{1}^{\left(2\right)}\right)$ and $\forall c_{2}\in I\left(a_{2}ns_{2}^{\left(2\right)}\right)$*,* we can conclude that ***c_1_***
***<***
***c_2_***.

If the sub-string ***ns_1_*** and ***ns_2_*** have the same prefix, there are two scenarios.

If $ns_{1}=ns_{1}^{\left(0,i\right)}ns_{1}^{\left(i\right)}$, $ns_{2}=ns_{2}^{\left(0,i\right)}ns_{2}^{\left(i\right)}$, and $ns_{1}^{\left(0,i\right)}=ns_{2}^{\left(0,i\right)}$ is the same prefix, *I (*$ns_{1}^{\left(0,i\right)})=I(ns_{2}^{\left(0,i\right)}$). Since $ns_{1}^{\left(i\right)}$ and $ns_{2}^{\left(i\right)}$ have different prefixes, and we proofed the order-preserving above. Therefore, in this scenario, Algorithm 2 is order-preserving as well.

If $ns_{1}=ns_{1}^{\left(0,i\right)}ns_{1}^{\left(i\right)}$, $ns_{2}=ns_{2}^{\left(0,i\right)}\lambda$ and $ns_{1}^{\left(0,i\right)}=ns_{2}^{\left(0,i\right)}$ is the same prefix *I (*$ns_{1}^{\left(0,i\right)})=I\;(ns_{2}^{\left(0,i\right)}$). According to the definition of *λ* node, the value of *λ* node is the smallest, so all the value in sub-interval of *λ* node are smaller than the value in the sub-interval of $ns_{1}^{\left(i\right)}$. Therefore, $\forall c_{1}\in I\left(ns_{1}^{\left(0,i\right)}ns_{1}^{\left(i\right)}\right)$ and $\forall c_{2}\in I\left(ns_{2}^{\left(0,i\right)}ns_{2}^{\left(i\right)}\right)$*,* we can conclude that ***c_1_***
***<***
***c_2_***.

Combing the above three scenarios, we can conclude that the Algorithm 2 is order-preserving.

### 2.4. The Encryption and Decryption of Character String in WRID-OPES

In the practical application of IOT sensing, encrypted objects usually include numbers, characters, strings, pictures, and other types. These data can be converted into hexadecimal digit strings through data encoding. Algorithm 3 provides a general order-preserving encryption supporting multiple types of data, and Algorithm 4 is a decryption algorithm for Algorithm 3.
**Algorithm 3:** The encryption scheme of WRID-OPESInputOutput*Userkey*, *DD**, N**, PS =* {*ps_1_,ps_2_,…ps_k_,**...,ps_λ_*} Ciphertext string: *CS*
Step1Using the pre-process method of character String in Section 2.1, encoding the {*ps_1_,ps_2_,…ps_k_,...,**ps_λ_*} into the numerical string set {*ns_1_,ns_2_,…ns_k_,...ns_λ_*} where the numerical value is hexadecimal and k is the number of strings.Step2Counting the occurrence times of each hexadecimal numerical character ***0*** to ***F*** in {*ns_1_,ns_2_,…ns_k_,...,ns_λ_*}. The occurrence times of ***λ*** is defined as the number of sub-string in {*ns_1_,ns_2_,…ns_k_,...,ns_λ_*}. Then we can obtain the occurrence frequency weight ***W = (w_λ_,w_0_,…,w_F_)***.Step3Generating the main seed seedp according to {*Userkey*, *DD, N*}.Step4Appending *W* to {*Userkey, DD, N*} as the decryption Keys.Step5Initialing the ciphertext set: ∅*→**CS.*Step6Constructing the weighted tree according to *W*.Step7If all strings be processed in {*ns_1_,ns_2_,…ns_k_,...,ns_λ_*}, then we output ***CS*** and the decryption Keys, and exit the algorithm, otherwise go to step 8.Step8Obtain an unprocessed string ***ns_k_*** from {*ns_1_,ns_2_,…ns_k_,…,ns_λ_*}. Using Algorithm 2 to encrypt this numerical string and obtain the ciphertext ***cs_i_.*** Appending cs_i_ to the ciphertext set ***CS***, e.g., ***CS = CS + {cs_i_}***Step9Go to step 6.

WRID-OPES is a symmetric encryption algorithm. The decryption likes the inverse process of encryption. However, in decryption, we cannot obtain the occurrence frequency weight ***W***. In Algorithm 4, we append ***W*** into the decryption key, which can be used to construct the weighted tree in decryption. In Algorithm 4, we detail the decryption processes of WRID-OPES.
**Algorithm 4:** The decryption scheme of WRID-OPESInputOutput*Userkey*, *DD**, N**, **CS, W***Plaintext string: *PS*Step1Obtaining ***W*** from the decryption Keys.Step2Generating the main seed ***seedp*** according to {*Userkey*, *DD, N**, W*}.Step3Initialing the plain text set: **∅****→*PS***.Step4Constructing the weighted tree according to ***W***Step5If all strings be processed in ***CS***, then we output *PS* and exit the algorithm.Step6Obtain the unprocessed string ***cs_i_*** from ***CS, CS = CS − {cs_i_}***Step7Using the Algorithm 2 to decrypt ***cs_i_*** and obtain the plaintext string ***ps_i_***, appending ***ps_i_*** to the plaintext string set *PS*, e.g., ***PS = PS + {ps_i_}.***

## 3. Experiments and Evaluations

In this section, a series of experiments is conducted to evaluate WRID-OPES. Firstly, our experiment environment is descripted. Then, the efficiency analysis results are presented. Finally, the security evaluations are described.

All the experiments are executed on a test machine with Intel (R) Core (TM) i7-7500U CPU @ 2.90 GHz and 8 GB RAM running Windows 10. We chose to use Python programming language to implement WRID-OPES.

### 3.1. The Efficiency Tests

In this section, we evaluate the efficiency of WRID-OPES from the aspects of computing time costs and storage space costs.

#### 3.1.1. The Tests of Computing Time Costs

In the following, we evaluate the computing time costs of WRID-OPES, compare with two other schemes: OPES [3] and LazySample [5]. Since all these three schemes are symmetrical encryption schemes, we only evaluated the encryption time costs. First, we analyzed the time complexity of these three schemes.

(1)The complexity of WRID-OPES.

WRID-OPES consists of three major parts, which are preprocess, WRID, and OPRIT. In the module of preprocess, the major work is to encode each character string data to the numerical data and count the weight for the numerical data. The time complexity of preprocess was O(N). In OPRIT, we obtain one character from the numerical string and compute the sub-interval by WRID. WRID consists of two major steps, which are constructing the weight tree and dividing the interval. The time complexity for these two steps are both O(logM). The time complexity of OPRIT is O(NlogM). Since *M* is always a constant, for example *M* is 17 when the hexadecimal is used, the time complexity of WRID-OPES is O(N).

(2)The complexity of OPES.

We can easily analysis that the time complexity of OPES is O(N). Readers who are interested in analyzing OPES can reference [3].

(3)The complexity of LazySample.

The time complexity of LazySample is related with the definition domain of plain text and the range of ciphertext. In [5], authors announced that for the data (size was N), the average time complexity is O(NlogD).

Without losing generality, we defined the average encryption time for one character as following:

**Definition** **6.**
*The average encryption time **T_ch_** for one character is defined as the following:*


(3)$$T_{ch}=\frac{T_{init}+T_{enc}}{len}$$
where ***T_init_*** refers to the initial time of the encryption schemes; ***T_enc_*** refers to the encryption time; len refers to the length of input string.

Figure 4 shows the relationship between single character encryption time ***T_ch_*** with different string length ***len*** for three algorithms when the definition domain is [1,2048].

From Figure 4, we can observe that WRID-OPES and LazySample (Boldyreva in [4]) have better performance to process the long string. The main reason is that the initial time occupies a larger proportion of the total time in WRID-OPES and LazySample. When the length is larger than 10, the encryption time occupies a larger proportion in the total time, and ***T_ch_*** is tending towards stability in all schemes.

From Figure 4, we can observe that WRID-OPES has less time costs than OPES (Agrawal in [3]) and LazySample. Although LazySample and WRID-OPES both need to generate random numbers, the generation of weighted random tree in WRID-OPES are completed in the initialization, so the encryption time per character is the shorter than LazySample. In addition, because OPES includes two-level bucket mapping for each encryption, it has longer encoding time compared with the other two algorithms.

Figure 5 shows the ***T_ch_*** of three different encryption schemes with different size of definition domain when string length is 1000. From the Figure 5, we can easily observe that size of definition domain has little impact on the performance of WRID-OPES. While, with the increase in definition domain size, ***T_ch_*** of LazySample and OPES increases exponentially.

#### 3.1.2. The Tests of Storage Spaces

In this section, we use the message expansion rate (MER) to evaluate the storage space costs. The message expansion rate is defined as the ratio of ciphertext length to plaintext length.

In the experiment, the range of ciphertext is [0,10^18^); the size of bucket is 10^7^, where all the values are long integer. Table 3 shows the average MER of 1000 random samples tested in hexadecimal encoding and Unicode encoding.

According to Table 3, we can easily observe that the message expansion is related to the character type and coding approach. When using the same coding approach to process the same type characters, the message expansion remained fairly static, regardless of how many plain texts are processed.

### 3.2. The Security Evaluations of WRID-OPES

In this section, we verify the security of WRID-OPES when facing the attacks using exhaustive methods or statistical methods.

#### 3.2.1. The Security Analysis of WRID-OPES

Boldyreva et al. in [5] defined the security notion IND-OCPA where the adversary can only query the left-or-right encryption oracle with ordered plaintext pairs. An encryption scheme is secure under IND-OCPA if the advantage of an efficient adversary (probability to distinguish whether the returned ciphertexts are encrypted from the left or the right plaintexts) is negligible.

Xiao et al. in [21] declared that IND-OCPA is the highest security level of OPES. However, Teranishi et al. in [22] proved that the OPES is vulnerable to big jump attacks, unless its ciphertext space is an exponential in the size of the plaintext space.

In 2013, Popa et al. in [9] proposed an ideal ordered encryption scheme, mOPE, which has significant security, does not disclose any plaintext except “order”, and has reached the IND-OCPA security on non-exponential ciphertext-space.

Compared with the ideal OPES, the plaintext space of our scheme is the number 0–F, and the ciphertext space is an exponential size (such as 16^8^, 16^16^). Compared with the mOPE, our scheme uses a weighted random tree to calculate the weight and allocate the ciphertext, so it does not disclose any plaintext except “order”. Therefore, our scheme can achieve the IND-OCPA security on the exponential ciphertext space.

#### 3.2.2. The Analysis with Exhaustive Attacks

The encryption results of WRID-OPES are determined by both the password and weight. The password determined the main seed (for example, ***seedp*** in Algorithm 2) when generating the random number. The encryption results are different if different passwords are used. The weight determines the proportionality coefficient when randomly dividing the interval.

WRID-OPES implements the encryption with OPRIT as computing unit. The complexity of OPRIT determines whether WRID-OPES can defend against the exhaustive attacks.

On the one hand, we assume the password to be a long integer main seed with 64 bits. There are 2^64^ possible main seeds. On the other hand, for the ciphertext (named *chex*) presented by hexadecimal, 16 hexadecimal characters refers to be a long integer; for the English character string (named *Pen*), one English character is encoded to two hexadecimal characters. According to the definition of message expansion, we can estimate the average length of the ciphertext is *2 *** Len (**Pen**)/(Len (**chex**)/**16**)***, where ***Len (****str**)*** is used to calculate the length of “str”.

According to the experimental results shown in Table 3, we estimate that the average length of the ciphertext is 10. Moreover, according to the results in computing time cost tests, we can assume that 0.103 ms is needed to encrypt one character. Therefore, 2^63^ × 10 × 0.103 ms (about 3.01 × 10^8^ years) is needed to exhaustive traverse all the possible keys when the weight is fixed. If the weight is changeable, the computing time is much larger than the time when weight is fixed. Therefore, we can conclude that WRID-OPES is able to defense the exhaustive attacks.

#### 3.2.3. The Attacks with Statistical Analysis

Once collecting enough ciphertext, attackers can use statistical analysis to attack the encryption scheme. The statistical analysis attack is called the frequency-based attacks.

In the attacks, there are two kinds of frequency, which are the occurrence frequency and distribution frequency. On the one hand, if the attacks obtain the occurrence times (*n*) of plain text (*p*) and obtain the ciphertext whose occurrence times are n, he can deduce the relationship between the plain text and ciphertext, thus obtaining the key. On the other hand, for the order-preserving encryption scheme for the numerical data, attackers can stretch the definition domain of plain text and the range of ciphertext into the same interval, such as [0,1). Then attackers can deduce the relationship between plain text and ciphertext by comparing the density probability plot of plain text and cipher text.

WRID-OPES is designed for the character string data. Moreover, the definition of distance between character strings is different from the distance between numerical data; therefore, OPRIT is the vulnerable step for suffering from the frequency-based attacks. The attackers can deduct the weight by counting the distribution frequency of the encrypted sub-intervals in the definition domain. Moreover, in the previous section, we proved that the expectation of division ratio was similar to the given weight. Therefore, we can conclude that the frequency distribution of long integer encrypted value is similar to the provided distribution. To verify this assumption, we conducted the test by using 1 000 random character string whose length is 100. We used the counted weight of numerical characters and the random characters as the input of WRID. Then obtained the distribution results as shown in Figure 6.

To count the frequency distribution of the encrypted results, we divided the definition domain into 20 same size sub-intervals and counted the number of encrypted value which dropped in different sub-intervals. Moreover, the experimental results are shown in Figure 7.

From the results shown in Figure 7, we can easily observe that the distribution of the ciphertext is similar to the uniform distribution when using the weight value as the input of WRID. While using the random value, we could not obtain such a result. Therefore, using WRID-OPES, the ciphertext distribution obeys the uniform distribution, regardless of what weight and what input characters. We can conclude that WRID-OPES can conceal the weight of plain text effectively.

### 3.3. Discussion on Related Works

Through the above analysis and experimentations, we summarized the computing cost, storage cost, security level, and plaintext data types for several typical OPES, as shown in Table 4. In this table, the computing cost refers to the encryption time, the storage cost refers to the space expansion rate of ciphertext relative to plaintext, and the data types refers to integer, float, character, and string.

## 4. Conclusions

In this paper, WRID-OPES, an order-preserving encryption scheme based on random interval division in the cloud was proposed. WRID-OPES consists of four parts: The first one is the preprocess modular, which encodes the input character string data according to the occurrence frequency. Second, we proposed the weighted random interval division algorithm (WRID), which divides the interval randomly and guarantees that the expectation of division ratios is similar to the weight W. Third, based on WRID, we proposed the order-preserving random interval tree (OPRIT), which is an order-preserving interval division algorithm. Finally, we presented the encryption and decryption process of WRID-OPES. According to the analysis, we verified that WRID-OPES can resist the exhaustive methods and statistical methods (e.g., the frequency-based attacks and size-based attacks). Moreover, we conducted series of simulation experiments to evaluate the efficiency of WRID-OPES. According to the experimental results compared with OPES and LazySample, we can draw the conclusion that WRID-OPES has linear encryption time and constant ciphertext expansion rate. Further works include: optimizing the computational performance of WRID-OPES for matching the embedded terminals, designing the improved privacy protection algorithms for developing applications in wearable devices; implementing the health monitoring system with privacy protection and ciphertext comparison; and so on.

## Figures and Tables

**Figure 1 sensors-22-07950-f001:**
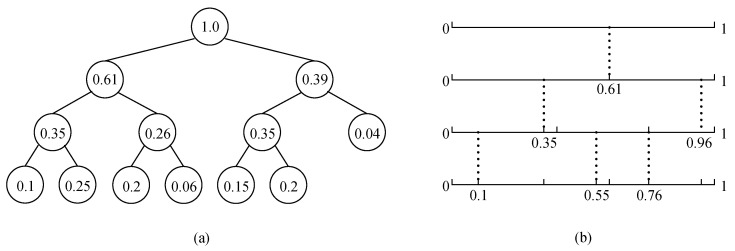
The relation between weighted tree and interval division. (**a**) is the random weighted tree, (**b**) is the weighted random interval division.

**Figure 2 sensors-22-07950-f002:**
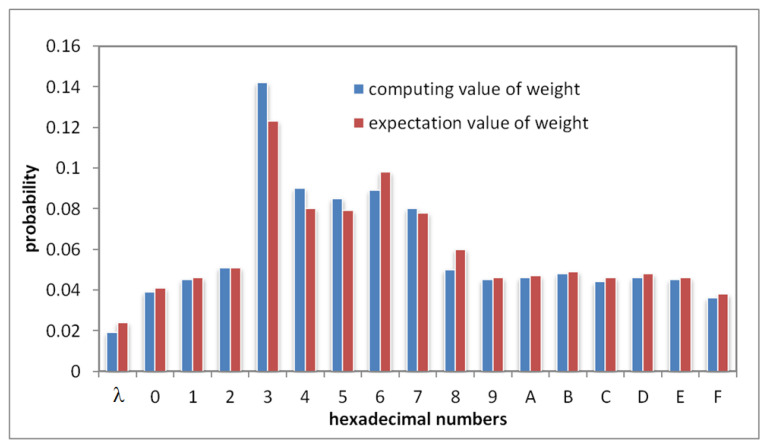
The comparison between the weight value and tests average value of the WRID algorithm.

**Figure 3 sensors-22-07950-f003:**
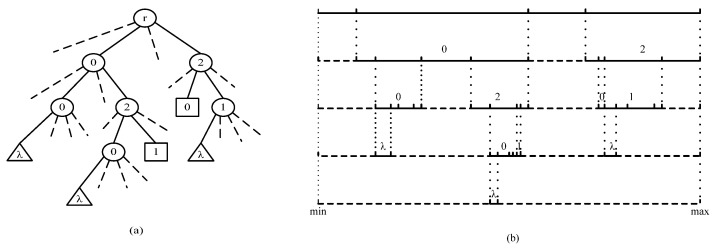
An example of OPRIT and the corresponding interval division results when M = 3. (**a**) is the encryption path, (**b**) is the interval division results.

**Figure 4 sensors-22-07950-f004:**
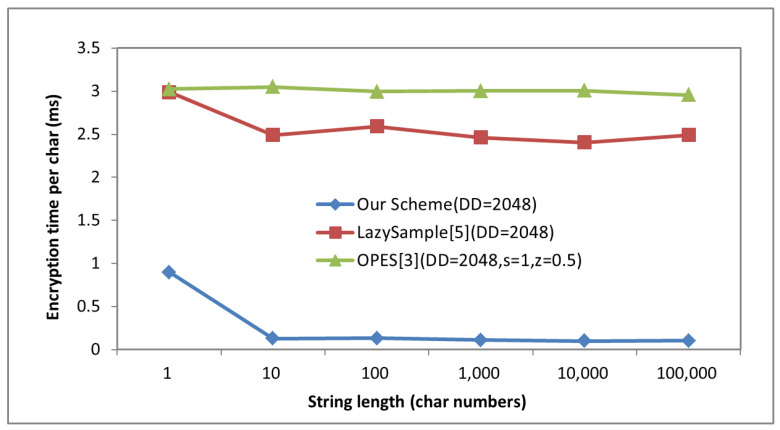
The encryption time of single character with different string length.

**Figure 5 sensors-22-07950-f005:**
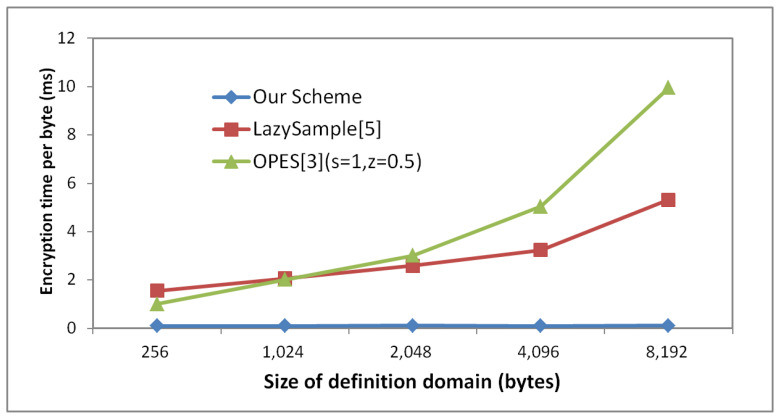
The encryption time with different size of definition domain.

**Figure 6 sensors-22-07950-f006:**
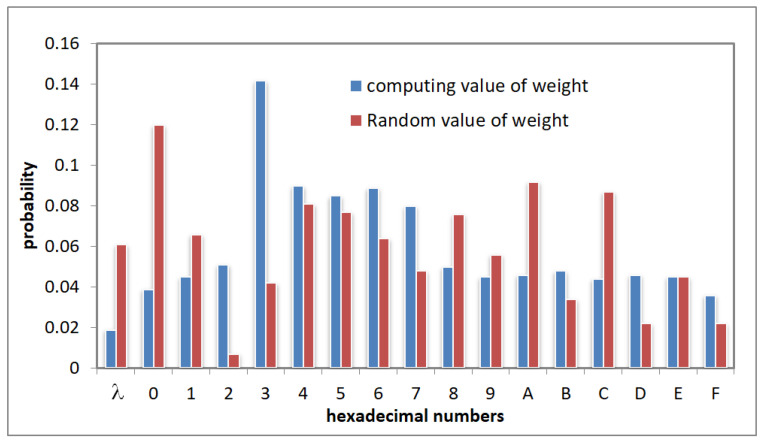
The distribution results of weight under different backgrounds.

**Figure 7 sensors-22-07950-f007:**
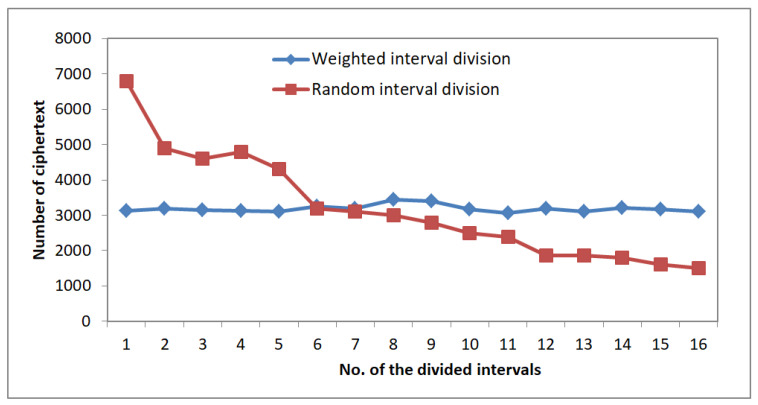
Total number of the ciphertext in weighted interval and random interval.

**Table 1 sensors-22-07950-t001:** The original UTF-16BE and changed UTF-16BE.

Character Type	Frequency	Original Code	Changed Code
Basic Latin characters	High	0000~007F	00~7F
Native characters and other types of characters	Low	0080~3FFF	80080~83FFF
Chinese, Japanese, and Korean (CJK) characters	High	4000~9FFF	9000~EFFF
Other East Asian characters	Low	A000~FFFF	FA000~FFFFF

**Table 2 sensors-22-07950-t002:** The randomness tests of WRID algorithm.

ID of Sub-Intervals	Correlation Coefficient *R*	Standard Deviation *σ* (10^8^)	Expectation *E**(X)* (10^8^)	Variation Coefficient *V_σ_* (%)
0	/	7.9818	3.9719	200.96
1	0.0449	5.7371	4.1860	137.05
2	0.0195	6.1134	4.5972	132.98
3	0.0061	4.7926	4.7304	101.32
4	−0.0133	10.1264	11.5040	88.03
5	0.0228	10.1896	9.2981	109.59
6	0.0330	6.8511	7.6025	90.12
7	0.0403	9.9952	8.9188	112.07
8	−0.0034	9.3557	7.6329	122.57
9	−0.0145	5.5705	4.8888	113.94
10	−0.0163	6.6836	3.9849	167.72
11	−0.0135	7.2979	5.2302	139.53
12	−0.0185	6.4425	5.0115	128.56
13	−0.0275	4.9870	4.9973	99.79
14	−0.0315	5.7014	4.8240	118.19
15	−0.0583	7.0227	5.2525	133.70
16	−0.0455	4.5780	3.3692	135.88

**Table 3 sensors-22-07950-t003:** The results for message expansion.

Character Type	Plaintext Length	Ciphertext Length in Hexadecimal	MER in Hexadecimal Encoding	Ciphertext Length in Unicode	MER in Unicode Encoding
English	20	63	3.15	15	0.75
English	100	282	2.82	70	0.7
English	1024	2812	2.75	696	0.68
Chinese	20	145	7.25	36	1.8
Chinese	100	694	6.94	174	1.74
Chinese	1024	7026	6.86	1775	1.73

**Table 4 sensors-22-07950-t004:** The results for message expansion.

Scheme	Computing Costs	Storage Costs	Security Level	Data Types
Agrawal04 [3]	Medium	Middle	Low	Integer
Boldyreval09 [4]	High	Low	Medium	Integer
Liu13 [7]	Low	Middle	Low	Integer, Float
Popa13 [9]	High	High	High	All data Types
Liu16 [14]	Middle	Middle	Middle	Integer, Float
Our Scheme	Low	Middle	High	All data Types

## Data Availability

The research data including programs and test codes of this paper can be shared from the following email: guiruowei@163.com.

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
