# Peer review of "An Order-Preserving Encryption Scheme Based on Weighted Random Interval Division for Ciphertext Comparison in Wearable Systems"

_sensors, 2022, doi:10.3390/s22207950_

Round 1

Reviewer 1 Report

The manuscript under review aims to propose an order preserving encryption scheme based on weighted random interval division. This paper provides an interesting result that realizes the order preserving encryption of multiple types of data with linear encryption time, small ciphertext expansion rate, and several attack resistances. However, there exist some questions and deficiencies. In the following, I summarize several points which should be improved.

1.    Introduction can be improved. The detailed related researches can move to Discussion section and I suggest to briefly introduce them here.

2.    I suggest to explicitly stated the proposition regarding to Proof in Line 329.

3.    The Discussion section can be added and improved. I suggest to list the comparison of several distinct schemes in time and storage after their detailed descriptions.

4.    I suggest to use figures with higher resolution in all Figures.

Author Response

Thank you for the positive comments and suggestions. We have revised our paper.  The following is the reply to your suggestions.

Point 1: Introduction can be improved. The detailed related researches can move to Discussion section and I suggest to briefly introduce them here.

Response 1: We simplified the content of "related researches" in "Introduction", and moved the details of "related researches" to the newly added Section “1.1. Related works".

Point 2: I suggest to explicitly stated the proposition regarding to Proof in Line 329.

Response 2: We have explicitly stated the description before Proof in line 329 of previous version as a Theorem.

Point 3: The Discussion section can be added and improved. I suggest to list the comparison of several distinct schemes in time and storage after their detailed descriptions.

Response 3: We have revised the structure of this paper, and added two Sections "1.1. Related works" and “3.3. Discussion on related works”. In this Section 1.1, we described the detailed design ideas of several typical OPE schemes, and then presented the comparison table of these schemes in time, storage and security in Section 3.3.

Point 4: I suggest to use figures with higher resolution in all Figures.

Response 4: We revised the unclear pictures in the paper (for example, Figure 2-7), and improved the resolution of these pictures.

Revised manuscript sees the attachment.

Reviewer 2 Report

This article proposes an order-preserving encryption scheme based on weighted random interval division for ciphertext comparison in wearable systems. The paper reflects substantial work and a solid  PoC.

The paper is not well organized. The authors should address the recommendations I have noted in their submitted article  to re-organize it (Please refer to my comments on the document.)

The authors should have a section in their introduction about their proposed method’s contributions. And mention their plans for future work at the end of their conclusion.

Author Response

Thank you for the positive comments and suggestions. We have revised our paper. The following is the reply to your suggestions.

Point 1: The paper is not well organized. The authors should address the recommendations I have noted in their submitted article to re-organize it (Please refer to my comments on the document.)

Response 1: We revised the structure of this paper, added the description of the main contributions of this paper in the Introduction as Section 1.2, and pointed out the further work in the conclusion. Future, according to the reviewer's comments, we adjusted Section 3.1 to Section 2.2 to illustrate the randomness of the WRID algorithm and the expectations of interval division percentage in WRID.

Point 2: The authors should have a section in their introduction about their proposed method’s contributions. And mention their plans for future work at the end of their conclusion.

Response 2: We revised the structure of this paper, added the description of the main contributions of this paper in the Introduction (see Section 1.2), and pointed out the further work in the conclusion. The further works include: optimizing the computational performance of WRID-OPES for matching the embedded terminals, designing the improved privacy protection algorithms for developing applications in wearable devices; implementing the health monitoring system with privacy protection and cipher-text comparison, and so on.

Revised manuscript sees the attachment.

Reviewer 3 Report

Indeed privacy of health data aggregation, comparative data analysis under privacy protection still face significant challenges. I agree that order-preserving encryption is an effective scheme to achieve private data retrieval and comparison, and that existing order-preserving encryption algorithms are targeted at integer data or single characters for the most part. The need to build a lightweight order-preserving encryption scheme that supports multiple data types, such as integers, floating numbers, and strings, is well justified.

The authors present an order-preserving encryption scheme (WRID-OPES) based on weighted random interval division (WRID), which can perform order-preserving encryption of multiple data types. A number of interesting steps are performed in the proposed process. A security analysis is also performed with the goal of reaching IND-OCPA security (indistinguishability under plaintext ordered selection attack) on static data sets, can resist attacks using exhaustive methods and statistical methods, and has a linear encryption time and small ciphertext expansion rate.

The article is well structured. The problem is well introduced and the design of WRID-OPES is well defined and the evaluations performed with the experiments seem to demonstrate the feasibility of the proposal.

Although there are certain aspects that could be improved, some figures such as 5,6, 7... appear blurred, it is recommended to improve the quality of the figures. With respect to future work, it seems to me that it would be interesting to describe a solution such as the one presented as a security pattern as in "Development of Applications Based on Security Patterns".  On the other hand, I believe that this work is related to "Security-enhanced ambient assisted living supporting school activities during hospitalisation", at least in the contextualization of the special issue of sensors.

Author Response

Thank you for the positive comments and suggestions. We have revised our paper. The following is the reply to your suggestions.

Point 1: Although there are certain aspects that could be improved, some figures such as 5,6, 7... appear blurred, it is recommended to improve the quality of the figures.

Response 1: We revised the unclear pictures in the paper (for example, Figure 2-7), and improved the resolution of these pictures.

Point 2: With respect to future work, it seems to me that it would be interesting to describe a solution such as the one presented as a security pattern as in "Development of Applications Based on Security Patterns".  On the other hand, I believe that this work is related to "Security-enhanced ambient assisted living supporting school activities during hospitalisation", at least in the contextualization of the special issue of sensors.

Response 2: Further work is added as shown: optimizing the computational performance of WRID-OPES for matching the embedded terminals, designing the improved privacy protection algorithms for developing applications in wearable devices; implementing the health monitoring system with privacy protection and cipher-text comparison, and so on. Thank the reviewer for very good application suggestions. In the future work, we will continue to reference these suggestions, and implement the application demonstration with OPES proposed in this paper.